# Effects of Subsurface Drainage on Soil Salinity and Groundwater Table in Drip Irrigated Cotton Fields in Oasis Regions of Tarim Basin

Yuhui Yang [1,2], Dongwei Li [3], Weixiong Huang [1,4], Xinguo Zhou [3], Zhaoyang Li [1,2,5], Xiaomei Dong [1] and Xingpeng Wang [1,2,5,*]

1   College of Hydraulic and Architectural Engineering, Tarim University, Alaer 843300, China
2   Key Laboratory of Modern Agricultural Engineering, Tarim University, Alaer 843300, China
3   Farmland Irrigation Research Institute, Chinese Academy of Agricultural Sciences, Xinxiang 453002, China
4   Hubei Key Laboratory of Yangtze Catchment Environmental Aquatic Science, School of Environmental Studies, China University of Geosciences, Wuhan 430078, China
5   Key Laboratory of Northwest Oasis Water-Saving Agriculture, Ministry of Agriculture and Rural Areas, Shihezi 832061, China
*   Correspondence: 120050073@taru.edu.cn

**Abstract:** As one global issue, soil salinization has caused soil degradation, thus affecting the sustainable development of irrigated agriculture. A two-year study was conducted in 2018 and 2019 to identify the effects of subsurface drainage spacing on soil salinity and groundwater level, the latter of which is in a high-water table in drip irrigation cotton fields in the Tarim Basin oasis in southern Xinjiang, China. Three subsurface drainage treatments, with a drain spacing of 10 m (W10), 20 m (W20), and 30 m (W30), respectively, and a drainage-absent treatment (CK), are tested. With CK, soil salinity in the 0–60 cm layer was accumulated within a year. In contrast, the subsurface drainage reduced the soil salinity at a leaching rate of 10–25%. When decreasing the drain spacing, it was found that the soil desalination rate increased significantly ($p < 0.05$) with good repeatability. Experimental results showed that the fitting equation of the soil salinity leaching curve could accurately describe the soil salinity leaching pattern of drip irrigation, and thus could be further used to inversely determine the theoretical drip irrigation leaching quota for those soils with different salinity degrees. As such, subsurface drainage could effectively control the groundwater table. Compared with CK, subsurface drainage deepened the groundwater table and mitigated the fluctuation of the groundwater level. These effects were strengthened by reducing the drain spacing. Correspondingly, the influence of the fluctuation of the groundwater table was reduced.

**Keywords:** subsurface drainage; soil salinity; groundwater table; drip irrigation; soil salinity leaching curve

## 1. Introduction

As one of the global issues, the soil salinization has resulted in soil degradation, thus affecting the sustainable development of irrigated agriculture [1,2]. The number of salt-affected agricultural lands increases annually, with more than 19.5% of the irrigated lands being affected globally [3].

Because of the high salinity of soil parent materials, shallow groundwater levels, strong evaporation, and less rainfall, the oasis farmlands of Tarim Basin in southern Xinjiang are experiencing particularly severe soil salinization. The film-mulched drip irrigation has been advocated and applied in the oasis of Tarim Basin for more than 20 years [4,5]. This technique is suitable for agricultural planting in arid areas [6]. However, studies have shown that long-term film-mulched drip irrigation would lead to secondary soil salinization, that is, applying longer drip irrigation results in more accumulation of soil

salinity [7]. Currently, more than 41.21% of the farmlands in southern Xinjiang are subjected to soil salinization and secondary salinization, resulting in low yields [8], and thus strongly restricting the agriculture sustainable development of southern Xinjiang. Therefore, it is necessary to conduct tests on the improvement of saline soils and promote practical applications of suitable technologies [9].

As a highly effective technique, subsurface drainage has been used to solve the problems of flooding and salinity hazards and is considered a fundamental measure for the improvement of saline soils [10,11]. Many scholars have carried out experiments on improving saline-alkali lands with subsurface drainage [12–14] and obtained encouraging results [15]. Subsurface drainage reduces the soil moisture and salinity environment by discharging excess soil water [16,17] and salts [18], and controlling the groundwater table [19], so as to improve the physiological growth index of plants [20] and increase crop yields [21,22].

Soils become salinized easily when the farmland's groundwater levels are shallow [23]. Feng et al. [24] believed that farmland groundwater tables could be effectively controlled by subsurface drainage. Wang et al. [25] carried out the experiment of applying subsurface drainage to film-mulched drip irrigation cotton field in Xinjiang and concluded that the soil salinity at the depth of 0–80 cm soil layer was decreased effectively; the water drainage amount, salt drainage amount, desalination ratio, and cotton seedling emergence rate reached optimum when the drain spacing was kept at 15 m. Zhang et al. [26] used the soil salinity leaching curves and fitting equations to simulate the soil salinity leaching desalination patterns and the leaching effects of flooding irrigation and concluded that the soil salinity leaching characteristics under three kinds of drain spacing (3, 6, 9 m) scenarios could be well simulated. The above research results confirm that subsurface drainage can reduce the degree of soil salinization by lowering the groundwater table and drip irrigation leaching. However, when overirrigation happens, it would increase the burden of subsurface drainage and enhance soil nutrient leaching [27] When drip irrigation is not enough, it could not achieve the expected soil leaching effect. Therefore, when film-mulched drip irrigations encounter shallow groundwater levels, the groundwater table distribution patterns and the soil salinity leaching characteristics of subsurface drainage farmlands need further investigation. When drip irrigation is used for leaching under the condition of subsurface drainage, the accurate prediction of irrigation amount needs to develop appropriate calculation methods.

In reference to the current research results, we carried out a two-year subsurface drainage experiment to improve soil salination in a saline cotton field that has received multi-year film-mulched drip irrigations in an oasis of Tarim Basin. Some subsurface drainage pipes were installed in local farmland with well-established cotton planting practices and agronomic management measures, and the resulting soil salinity and groundwater tables were measured. The main objectives of this study are to firstly explore the effects of the drainage drain spacing on soil salinity and irrigation leaching desalination rate of drip irrigation, and clarify the response relationship between groundwater table and the drain spacing; secondly, check the applicability of soil salinity leaching curves under tested conditions, propose the calculation methods of theoretical drip irrigation leaching quota of soil salinity leaching, and provide scientific guidance for accurate soil salinity leaching control under the condition of drip irrigation plus subsurface drainage.

## 2. Materials and Methods

### 2.1. Test Area

A cotton field, which has received film-mulched drip irrigation for a few years, was tested. The test field locates at the Fifth Company of the Sixteenth Regiment of Alar City, First Division of Xinjiang Production and Construction Corps. The geographical location, soil physical properties, and meteorological data of the test area are shown in Figures 1 and 2 and Table 1. The test area's hydrological and meteorological conditions and the soil physical properties' detection methods are described in [20].

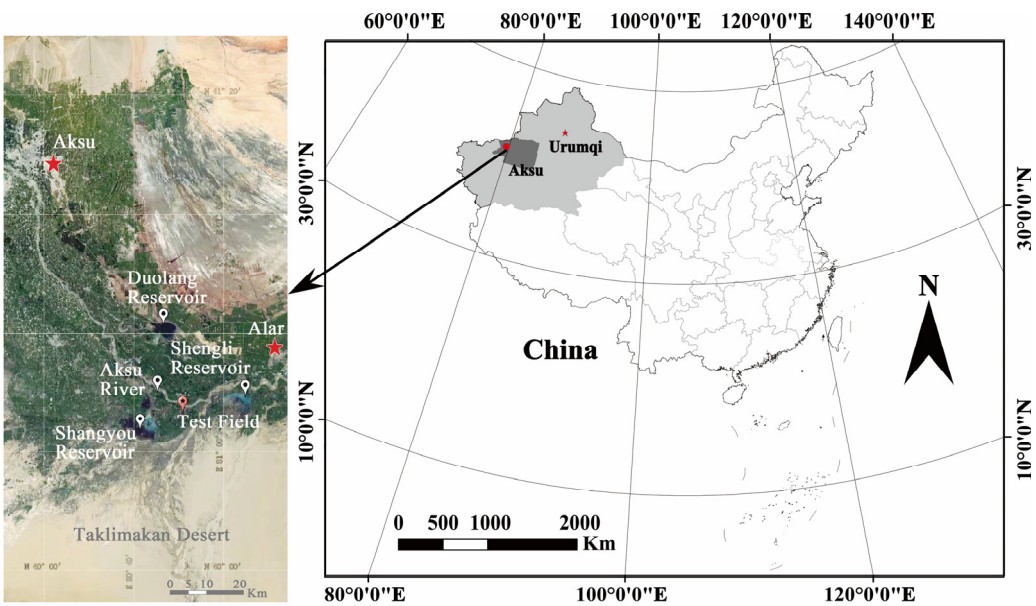

**Figure 1.** Geographical location of the test field.

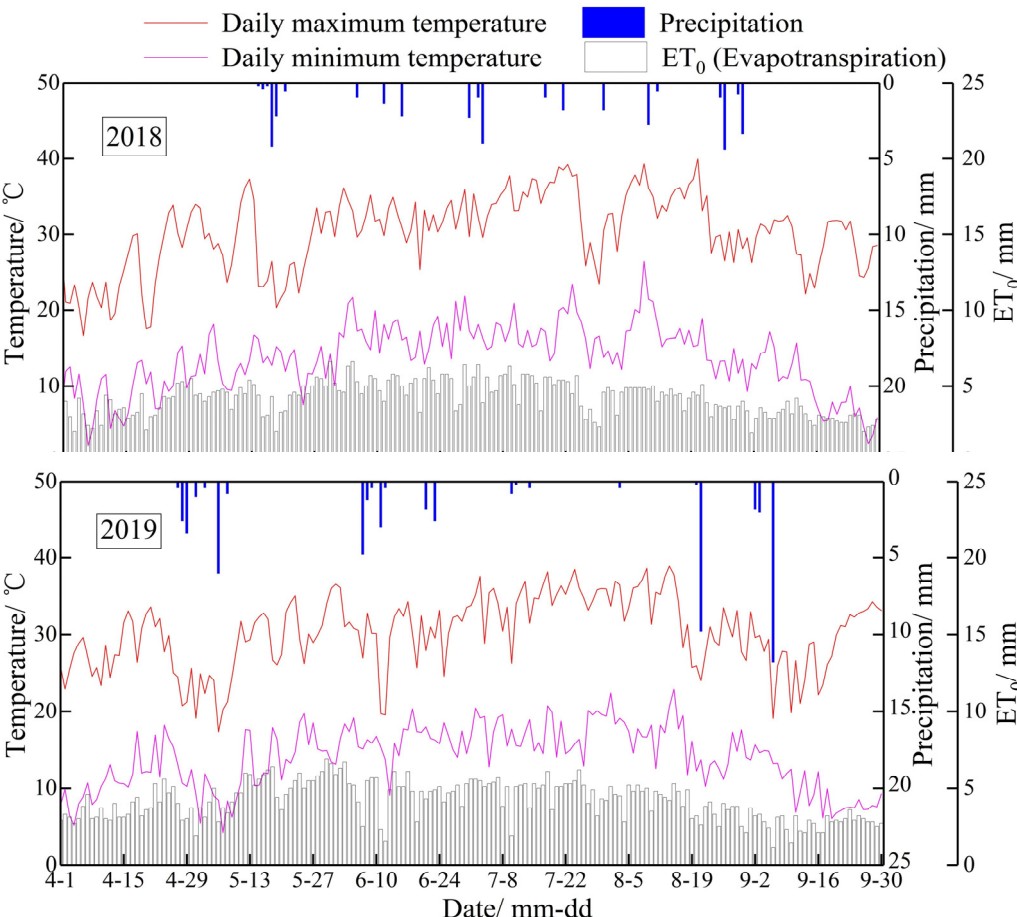

**Figure 2.** Meteorological data of the test area.

**Table 1.** Soil physical properties of the experimental field.

| Soil Depth, cm | | 0–20 | 20–40 | 40–60 | 60–80 | 80–100 |
|---|---|---|---|---|---|---|
| Soil bulk density, g/cm | | 1.51 | 1.48 | 1.43 | 1.34 | 1.36 |
| Saturated soil moisture, % | | 32.11 | 34.03 | 36.88 | 37.71 | 37.89 |
| Field water-holding capacity, % | | 26.25 | 27.06 | 30.83 | 32.31 | 33.68 |
| permeability coefficient, cm/d | | 11.2 | 8.8 | 8.1 | 8.6 | 7.9 |
| | Sand, % | 66.32 | 68.82 | 76.51 | 84.46 | 91.84 |
| Soil texture | Silt, % | 29.63 | 27.26 | 21.36 | 12.07 | 6.63 |
| | Clay, % | 4.05 | 3.92 | 2.13 | 3.47 | 1.53 |

*2.2. Test Design and Agronomic Management*

There are four treatments, which include three with a drain spacing of 10 m (W10), 20 m (W20), and 30 m (W30) at a burial depth of 1.1 m, and one as the control treatment without drainage (CK). The modes of subsurface pipe arrangement and cotton planting are shown in Figures 3 and 4, and the irrigation date and amount are shown in Table 2. The subsurface pipe laying construction and cotton agronomy management are described in [20].

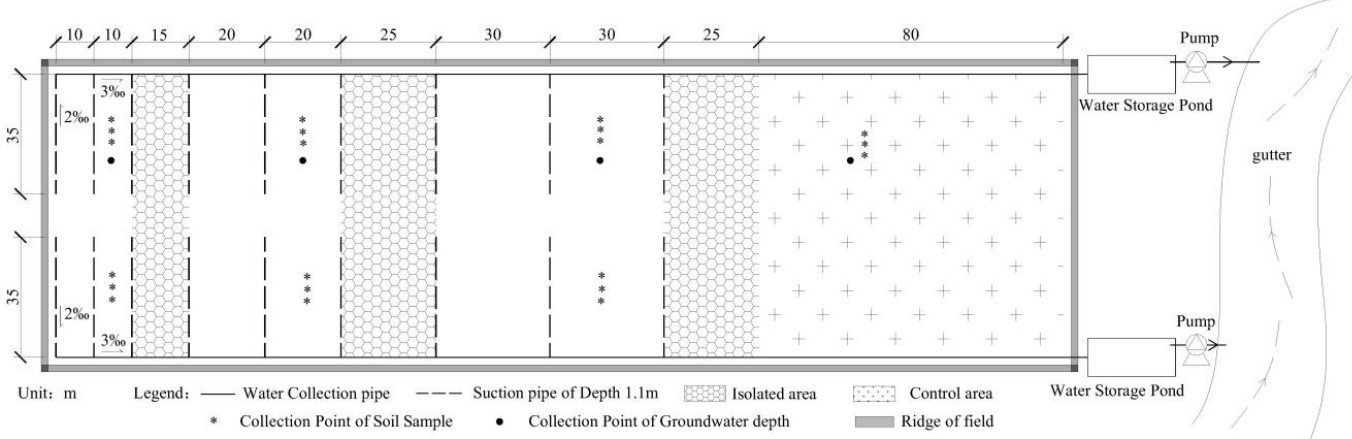

**Figure 3.** The drainage pipe layout and soil sampling point.

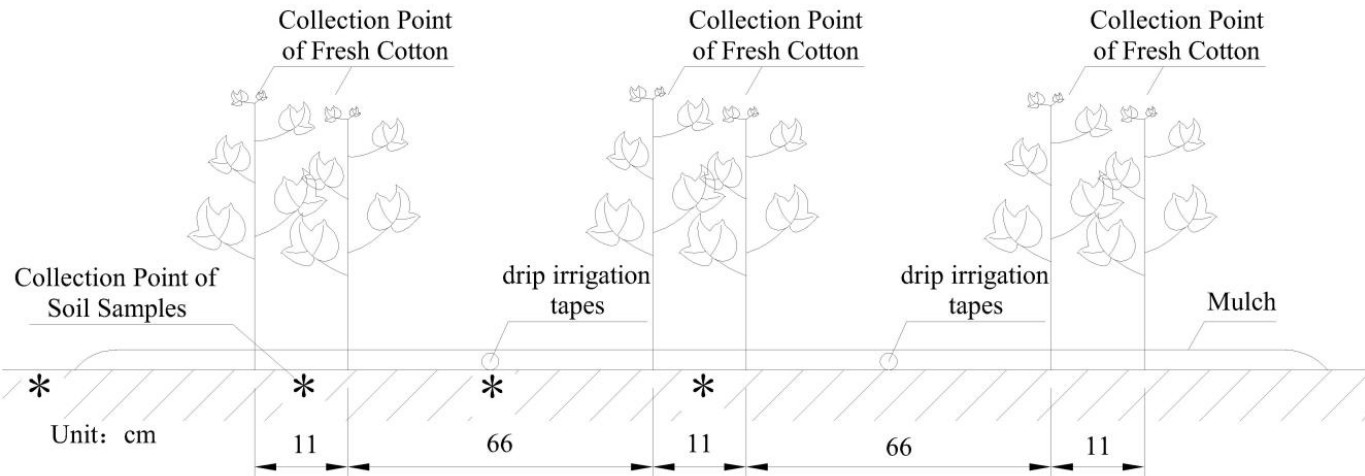

**Figure 4.** The planting layout and sampling sites.

**Table 2.** Irrigation scheme.

| Year | Irrigation Date (d-m) | Irrigation Quota (mm) |
|------|------------------------|------------------------|
| 2018 | 28-Jun | 37.5 |
|      | 11-Jul | 45 |
|      | 25-Jul | 45 |
| 2019 | 27-Jun | 37.5 |
|      | 13-Jul | 45 |
|      | 26-Jul | 45 |

*2.3. Data Collection*

The mean soil salinity of the 0–60 cm layer was calculated by a weighted average of the five soil layers (0–2 cm, 2–10 cm, 10–20 cm, 20–40 cm, and 40–60 cm) sampled. The determination method of soil salinity is described in [20].

Groundwater tables under the conditions of four treatments were measured by a Pressure Sensor (Gems, 2600BGA1019M3JA). The data were collected twice a day (Internet of Things data collection terminal, CZ-1). The groundwater table collection points are shown in Figure 3.

The soil leaching desalination rate was calculated based on the soil salinity difference before and after irrigation. The calculation formula is as follows:

$$D = \frac{C_2 - C_1}{C_1}$$ (1)

where $D$ is soil desalination efficiency before and after irrigation in %, $C_1$ is soil salinity before irrigation in g/kg, and $C_2$ is soil salinity after irrigation in g/kg.

The soil salinity leaching curve is a fitting equation constructed from soil salinity before and after irrigation and the irrigation leaching quota, reflecting the irrigation leaching effect and the soil desalination pattern [26]. The soil salinity leaching curve equation is as follows:

$$\frac{C_i - C_e}{C_0 - C_e} = a\left(\frac{D_w}{D_s}\right)^b$$ (2)

where $C_0$ is the soil salinity in the calculated soil layer before irrigation in g/kg, $C_i$ is the soil salinity in calculated soil layer after irrigation in g/kg, $C_e$ is the balanced soil salinity of irrigation leaching in g/kg, $D_w$ is the irrigation leaching quota in cm, $D_s$ is the thickness of calculation soil layer in cm, $a$ and $b$ are fitting parameters.

More details are described in one of our publications [20].

## 3. Results and Analyses

### 3.1. Effects of Drain Spacing on Soil Salinity Distribution and Desalination Efficiency

Six times of irrigation were carried out during the growth period from 2018 to 2019. The soil salinity distribution in cotton fields under drip irrigation and subsurface drainage is shown in Figure 5. Soil salinity distribution was in the order of CK > W30 > W20 > W10 before the irrigation of squaring stage in 2018, and the soil salinity decreased obviously after each irrigation. Water-carried salt moves up to the observation soil layer due to strong evapotranspiration after irrigation, which increases the soil salinity, thus each treatment presents a "wave-like" change pattern.

All the soil salinity of W10, W20, and W30 fluctuated slightly, ranging from 2.03–2.83, 2.48–3.49, and 2.85–3.93 g/kg in 2018, and 2.22–2.84, 2.96–3.72, and 3.34–4.38 g/kg in 2019, respectively. The soil salinity of CK and W30 decreased after irrigation. However, a stepwise increase and thus accumulation in soil salinity were observed throughout the whole year. Soil salinity distribution was in the order of CK > W30 > W20 > W10 after irrigation during the full-boll stage in 2019. In summary, these observations indicated that the soil salinity

decreased and fluctuated in a smaller range when the drain spacing decreased. The soil salinity of CK and W30 presented accumulation throughout the whole year.

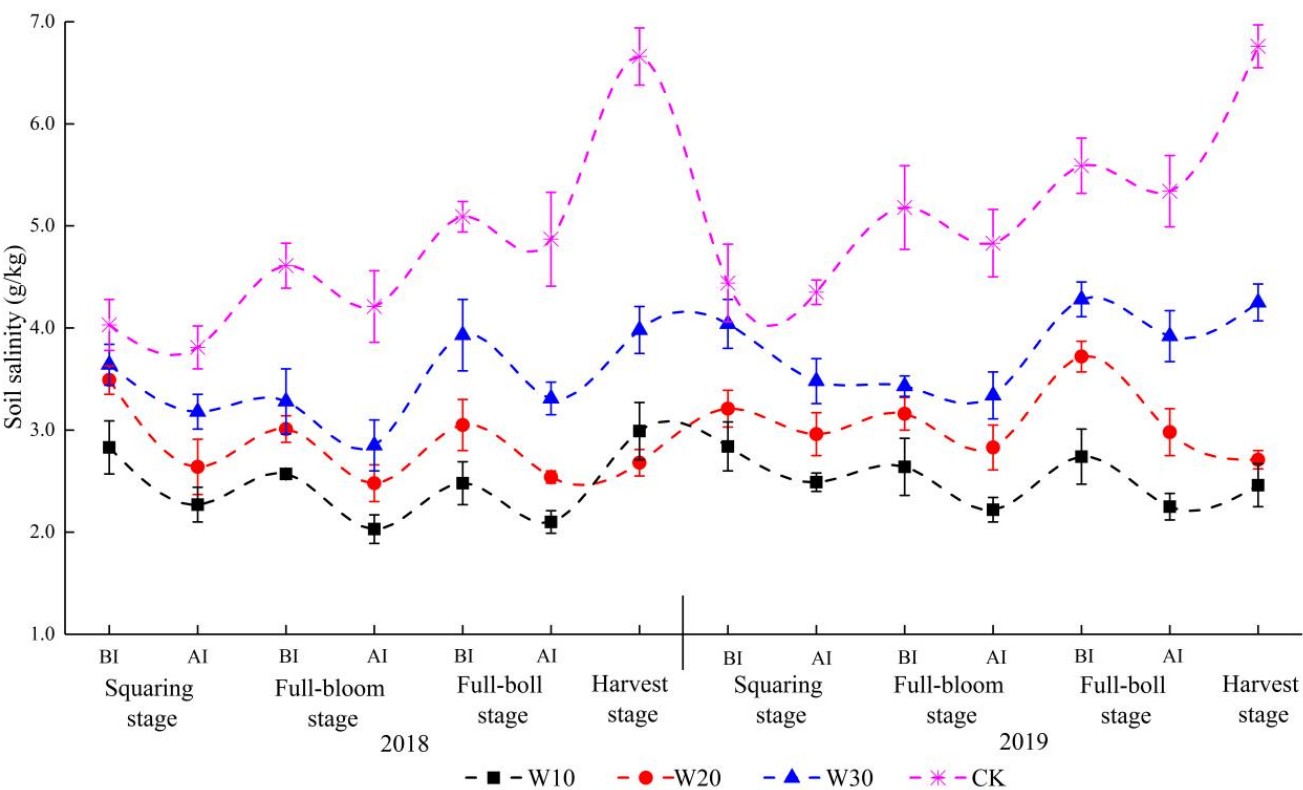

**Figure 5.** Soil salinity content before and after irrigation. W10, W20, and W30 mean the drain spacing of 10, 20, and 30 m, respectively. CK means no drainage. BI and AI mean before and after irrigation, respectively.

In order to analyze the leaching effect of drip irrigation and subsurface drainage during the cotton growth period quantitatively, the soil salinity leaching rate of each drip irrigation was calculated according to the average soil salinity of 0–60 cm soil layer before and after irrigation (Table 3). The soil salinity leaching rate of each irrigation in subsurface drainage was between 10% and 25% due to the small amount of drip irrigation, however, they were higher than the values of no drainage, with significant differences ($p < 0.05$). These observations indicate that the soil salinity leaching effect of drip irrigation was not obvious without drainage, and subsurface drainage improved the soil salinity leaching rate of drip irrigation significantly ($p < 0.05$).

*3.2. Soil Salinity Leaching Curves*

According to the experimental design, four calculated soil layer depths ($D_s$) were selected, which were 0–10 cm, 0–20 cm, 0–40 cm and 0–60 cm, respectively. The drip irrigation leaching quota ($D_w$) was consistent with the irrigation quota in squaring, full-bloom, and full-boll period, which were 3.75 cm, 4.5 cm, and 4.5 cm, respectively. The balanced soil salinity of irrigation leaching ($C_e$) refers to the surface soil (0–2 cm soil layer) salinity after leaching. Generally, $C_e$ is the product of the surface soil saturated moisture (32.11%) and the mineralization degree of irrigation water (1.00 g/L) [26]. The fitting parameters and soil salinity leaching curve are shown in Figure 6.

**Table 3.** Soil salinity leaching rate of drip irrigation, %.

| Year | Treatment | Squaring Period | Full-Bloom Period | Full-Boll Period |
|------|-----------|-----------------|-------------------|------------------|
| 2018 | W10 | 19.52 ± 1.26 [b] | 20.86 ± 1.02 [a] | 15.34 ± 0.62 [a] |
| | W20 | 24.28 ± 1.50 [a] | 17.65 ± 0.72 [b] | 16.55 ± 1.32 [a] |
| | W30 | 12.53 ± 0.90 [c] | 13.25 ± 0.50 [c] | 15.88 ± 0.94 [a] |
| | CK | 5.43 ± 0.67 [d] | 8.69 ± 1.04 [d] | 4.33 ± 1.20 [b] |
| 2019 | W10 | 12.35 ± 1.16 [b] | 15.81 ± 1.23 [a] | 17.99 ± 0.39 [a] |
| | W20 | 7.72 ± 0.34 [c] | 10.39 ± 0.72 [b] | 10.10 ± 1.15 [b] |
| | W30 | 14.01 ± 0.13 [a] | 2.68 ± 0.29 [d] | 8.44 ± 0.32 [b] |
| | CK | 2.02 ± 0.91 [d] | 6.77 ± 1.46 [c] | 4.38 ± 1.30 [c] |

Note: ± indicates the error value. W10, W20, and W30 mean the drain spacing of 10, 20, and 30 m, respectively. CK means no drainage. Different lowercase letters indicate values that are significantly different ($p < 0.05$) for comparisons within same year, index, and list.

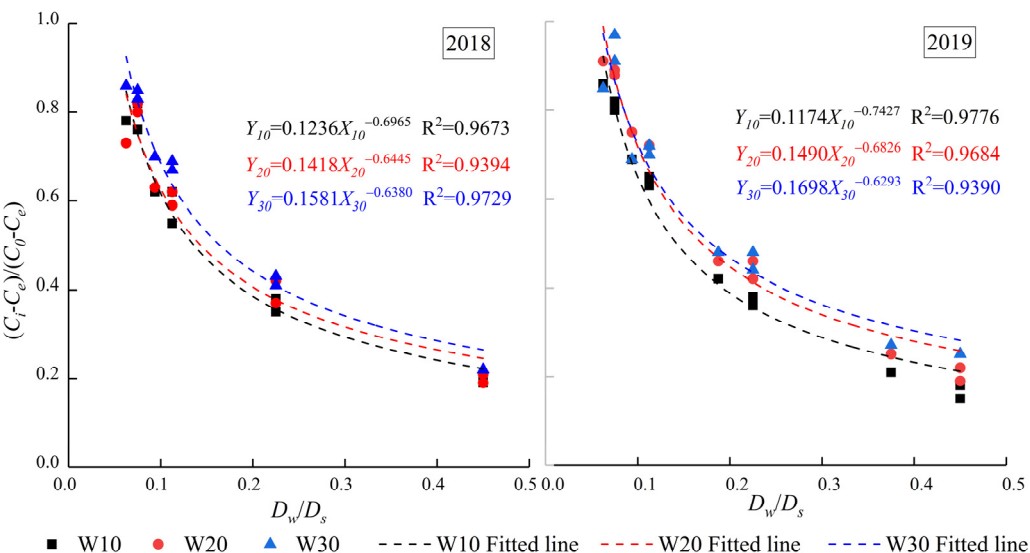

**Figure 6.** Soil salinity leaching curve. W10, W20, and W30 mean the drain spacing of 10, 20, and 30 m, respectively. CK means no drainage.

The determination coefficients ($R^2$) of soil salinity leaching curves in each treatment are higher than 0.93, indicating that soil salinity leaching curves correctly describe the soil salinity leaching regular of drip irrigation. The soil salinity leaching curve fitting parameters "a" are in the order of W30 > W20 > W10, indicating that the smaller the spacing of pipes are, the better the irrigation leaching effect results, which was consistent with the regular of "soil salinity decreased when the drain spacing decreased".

### 3.3. The Influence of Subsurface Drainage Drain Spacing on Groundwater Table

In the test cotton filed, the groundwater table variation trend under the condition of subsurface drainage was similar to CK (Figure 7). The average groundwater table of W10, W20, W30, and CK (Table 4) were 1.12 m, 1.00 m, 0.84 m, and 0.68 m in 2018, and 1.03 m, 0.92 m, 0.79 m and 0.64 m in 2019, respectively, indicating that the groundwater table were in the order of W10 > W20 > W30 > CK ($p < 0.05$). The groundwater table fluctuation ranges of W10, W20, W30, and CK were 1.00–1.20 m, 0.8–1.15 m, 0.55–1.10 m, and 0.3–0.95 m in 2018, and 0.95–1.1 m, 0.75–1.05 m, 0.65–1.00 m, and 0.35–0.9 m in 2019, respectively, indicating that the groundwater table fluctuation ranges were in the order of CK > W30 > W20 > W10. These observations indicated that without drainage the groundwater table was shallow

and displayed more fluctuation. The groundwater table reduced significantly ($p < 0.05$) and displayed less fluctuation with decrease in drain spacing.

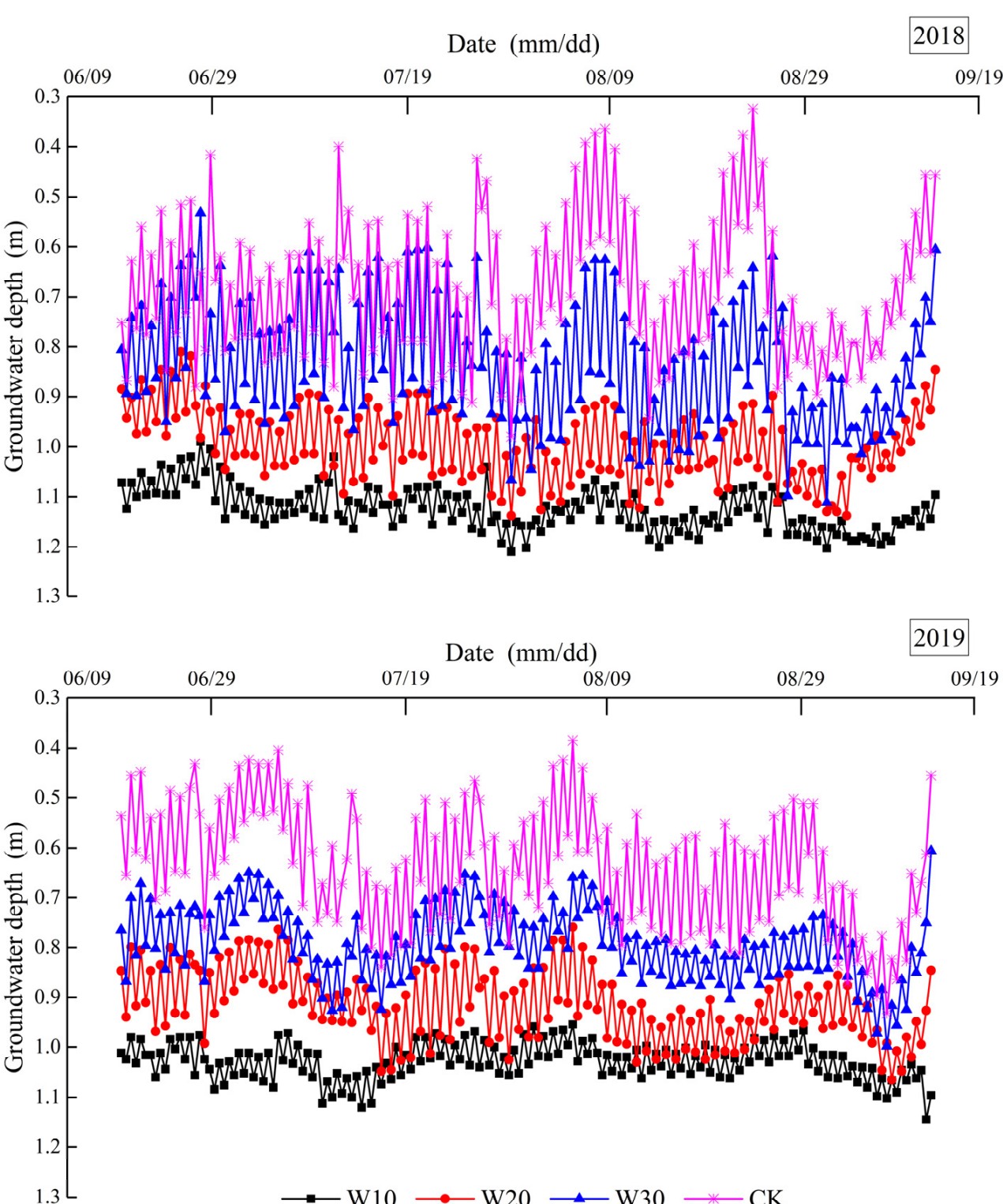

**Figure 7.** Groundwater table. W10, W20, and W30 mean the drain spacing of 10, 20, and 30 m, respectively. CK means no drainage.

**Table 4.** Groundwater table in 2-year growth period.

| Year | Treatment | June | July | August | September | Average |
|------|-----------|------|------|--------|-----------|---------|
| 2018 | W10 | 1.063 [a] | 1.121 [a] | 1.138 [a] | 1.161 [a] | 1.12 [a] |
|      | W20 | 0.914 [b] | 0.995 [b] | 1.025 [b] | 0.999 [ab] | 1.00 [b] |
|      | W30 | 0.772 [c] | 0.812 [c] | 0.866 [c] | 0.882 [b] | 0.84 [c] |
|      | CK | 0.678 [d] | 0.706 [d] | 0.657 [d] | 0.703 [c] | 0.68 [d] |
| 2019 | W10 | 1.018 [a] | 1.034 [a] | 1.015 [a] | 1.058 [a] | 1.03 [a] |
|      | W20 | 0.878 [b] | 0.898 [b] | 0.933 [b] | 0.967 [b] | 0.92 [b] |
|      | W30 | 0.766 [c] | 0.774 [c] | 0.800 [c] | 0.869 [c] | 0.79 [c] |
|      | CK | 0.573 [d] | 0.614 [d] | 0.637 [d] | 0.787 [d] | 0.64 [d] |

Note: Different lowercase letters indicate values that are significantly different ($p < 0.05$) for comparisons within the same year, index, and list. Same below.

## 4. Discussion

Subsurface drainage mainly suppresses the increase of soil salinity by lowering groundwater levels and draining excess water from the soil [28], thus the soil salinity under the condition of subsurface drainage would be significantly reduced compared with no drainage [29,30]. Our results showed that under the condition of subsurface drainage, the 0–60 cm soil layer desalination leaching ratio of drip irrigation was found to be 10–25%, a similar value to what was observed by He et al. [31]. A decrease in drain spacing resulted in a decrease in soil salinity ($p < 0.05$) but an increase in soil desalination leaching ratio. Without drainage, the soil salinity leaching effect was not obvious when applying drip irrigation. Furthermore, the soil salinity accumulation occurred within and between years, consistent with what was found by Wang et al. [13].

A fitting equation of soil salinity leaching curve can be constructed by soil salinity before and after irrigation and the irrigation leaching quota, reflecting the irrigation leaching effects and the soil desalination patterns [26,32]. Zhang et al. [26] constructed a soil salinity leaching equation under the condition of subsurface drainage plus flood irrigation. The results showed that the fitting equation of the soil salinity leaching curve could well describe the soil salinity leaching characteristics of flood irrigation. The soil salinity leaching effect increased with decreasing drain spacing, similar to what we found in this study.

According to the fitting equation of the soil salinity leaching curve in each treatment (Figure 6), different drip irrigation leaching scenarios (the soil salinity before leaching $C_0$ is 5.0 g/kg and 3.0 g/kg, respectively. The soil salinity after leaching $C_i$ is 2.0 g/kg, 1.5 g/kg, and 1.0 g/kg, respectively) were set, and the theoretical drip irrigation leaching quota (mm) of 0–60 cm soil layer was calculated accordingly. The calculation results are shown in Table 5.

As shown in Table 5, firstly, when the soil salinity before leaching $C_0$ is consistent, the difference in soil salinity after leaching $C_i$ under different scenarios is not proportional to the difference in theoretical drip irrigation leaching quota. For example, the data of 2019 show that under the condition of 10 m drain spacing, the theoretical drip irrigation leaching quota was calculated to be 62.9 mm and 212.6 mm, respectively, with the same $C_0$ (3.0 g/kg) but different $C_i$ (2.0 g/kg and 1.0 g/kg, respectively). It can be deduced that the theoretical drip irrigation leaching quota is 62.9 mm and 149.7 mm (equal to 212.6–62.9 mm), respectively, when the soil salinity was leached from 3.0 g/kg to 2.0 g/kg and from 2.0 g/kg to 1.0 g/kg. The theoretical drip irrigation leaching quota of the latter case is 2.38 times as large as that of the former case, indicating that the smaller the soil salinity is, the lower the irrigation leaching efficiency results. Secondly, when the soil salinity after leaching $C_i$ is kept consistent, the value of $C_i - C_0$ is not proportional to the difference in theoretical drip irrigation leaching quota. For example, in the data of 2019, the soil salinity difference leaching from 5.0 g/kg to 2.0 g/kg is 3.0 g/kg, and the theoretical drip irrigation leaching quota of 10 m drain spacing is 133.2 mm. In con-

trast, the soil salinity difference leaching from 3.0 g/kg to 2.0 g/kg is 1.0 g/kg, and the theoretical drip irrigation leaching quota in the same 10 m drain spacing is 62.9 mm. Theoretically, the theoretical drip irrigation leaching quota should be increased to a 3-fold value (62.9 mm × 3 = 188.7 mm) when the soil salinity leaching amount increases from 1.0 g/kg to 3.0 g/kg, but the actual calculation result is only 133.2 mm, indicating that the higher the soil salinity is, the better the drip irrigation leaching efficiency results. Thirdly, the theoretical drip irrigation leaching quota is reduced with decreasing drain spacing under every scenario, that is, W30 > W20 > W10 on each row, indicating that the smaller drain spacing saves more water at the same leaching efficiency. All these observations indicate that the subsurface drainage leaching efficiency is related to the soil salinity; for the soil with higher salinity, the drip irrigation leaching efficiency is better under the condition of the same quota; reducing the drain spacing can decrease the drip irrigation quota and save more water under the condition of the same leaching efficiency.

**Table 5.** Theoretical drip irrigation leaching quota under different scenarios.

| Year | Soil Salinity before Leaching, $C_0$ (g/kg) | Soil Salinity after Leaching, $C_i$ (g/kg) | Theoretical Drip Irrigation Leaching Quota (mm) | | |
|------|------|------|------|------|------|
| | | | W10 | W20 | W30 |
| 2018 | 5.0 | 2.0 | 129.8 | 142.0 | 168.1 |
| | | 1.5 | 215.6 | 245.6 | 290.8 |
| | | 1.0 | 475.7 | 577.7 | 684.0 |
| | 3.0 | 2.0 | 58.3 | 59.8 | 70.8 |
| | | 1.5 | 96.8 | 103.4 | 122.5 |
| | | 1.0 | 213.6 | 243.3 | 288.0 |
| 2019 | 5.0 | 2.0 | 133.2 | 165.5 | 182.6 |
| | | 1.5 | 214.4 | 277.6 | 320.1 |
| | | 1.0 | 450.3 | 622.5 | 768.6 |
| | 3.0 | 2.0 | 62.9 | 73.1 | 75.3 |
| | | 1.5 | 101.2 | 122.7 | 132.0 |
| | | 1.0 | 212.6 | 275.1 | 316.9 |

Farmland overirrigation results in soil nutrient loss and groundwater pollution [27], while insufficient irrigation is unable to achieve expected soil salinity leaching efficiency. Under the specific geological conditions of the test area, equation fitting is an effective approach to estimate the required drip irrigation quota to leach the soil salinity from different salinization degrees to a target value, thus avoiding farmland over or insufficient irrigation. Nevertheless, different regions normally have different salt leaching curve equations due to the different conditions of soil texture, irrigation water quality, etc. Therefore, it is important to determine the soil salinity leaching curves and generate fitting equations in different agricultural regions.

Subsurface drainage can effectively control the groundwater table [24]. Our test results showed that when compared with CK, the subsurface drainage could reduce the groundwater table significantly ($p < 0.05$), and a reduction in drain spacing had the same effect ($p < 0.05$). This is due to the formation of a parabola-shaped water table situated between two subsurface pipes [33].

To analyze the correlation of groundwater tables between the condition of drain spacing and that of no drainage, the subsurface drainage groundwater table data, and the no drainage (CK) groundwater table data were used to draw the scattered point graphs and fit with linear correlation equations. As shown in Figure 8, the distribution pattern of each group scattered points is obvious and the boundary is clear. In each fitting equation, the

slope reflects the fluctuation range of the groundwater table, the constant term reflects the groundwater table, and the $R^2$ value reflects the influence of no drainage groundwater table on the subsurface drainage groundwater table. According to the groundwater table fitting equations, the slopes of the fitting lines are in the order of W30 > W20 > W10, and the constant terms are in the order of W10 > W20 > W30, indicating that both the groundwater table and the fluctuation range reduced with decreasing drain spacing, which is consistent with the conclusion from Figure 7. The R2 values are in the order of W30 > W20 > W10, indicating that due to the influence of groundwater table fluctuation in no drainage area, the response of the groundwater table in W30 is the most obvious, followed by W20, and W10 is the least obvious. The drain spacing is a key parameter to affect the drainage capacity. Subsurface drainage amount increased significantly with decreasing drain spacing [25,29]. This is the key factor for us to obtain the following results: decreasing drain spacing results in lower groundwater tables and fewer fluctuation ranges, and less influence of the groundwater fluctuation in no drainage area on them.

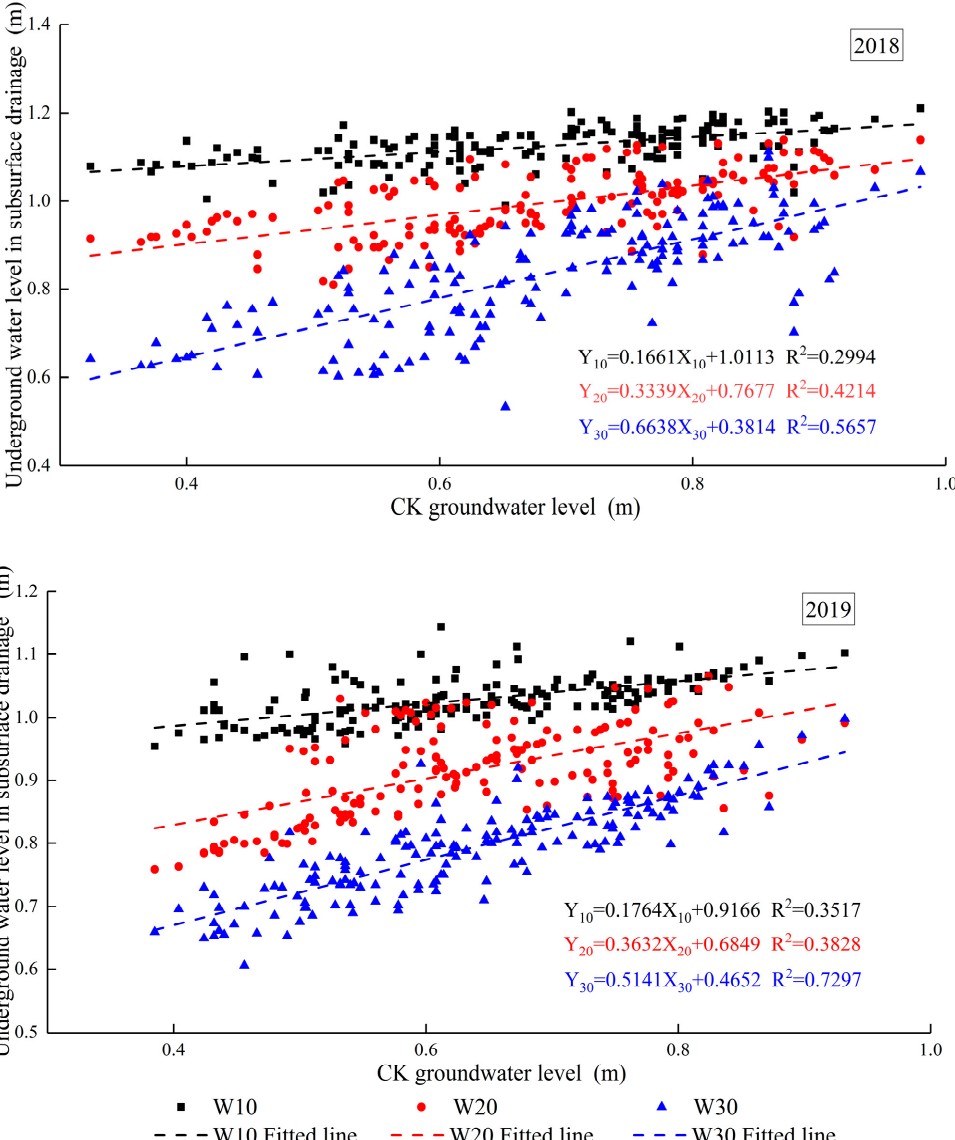

**Figure 8.** Response relationship between groundwater tables of subsurface drainage and groundwater table of CK. W10, W20, and W30 mean the drain spacing of 10, 20, and 30 m, respectively. CK means no drainage.

Shallow saline groundwater is a critical factor resulting in farmland waterlogging and soil salinization in arid and semi-arid areas [34,35]. Maintaining a deep groundwater table is an effective measure to control soil salinity and improve crop yields for saline farmland [36]. The groundwater quality is poor in the test area (the salinity is about 5.0 g/L), and the groundwater table in no drainage area is affected by the surrounding hydrological conditions (Figure 1: subsurface drainage not only reduced the groundwater table in test area, but also the influence of the groundwater table fluctuation in no drainage area on rising groundwater table in drainage area. With the reduction of the drain spacing, the saline groundwater table was reduced significantly ($p < 0.05$), and the influence from the groundwater table fluctuation in no drainage area decreased, too. This is also an important way for subsurface drainage to effectively improve the saline soil in the test area.

## 5. Conclusions

Experiments about soil improvement and groundwater table regulation under subsurface drainage were performed from 2018 to 2019 in a saline cotton field with a shallow groundwater table in the Tarim Basin oasis. The main conclusions are as follows:

(1) Compared with CK, the subsurface drainage decreased the soil salinity. Soil salinity decreased and fluctuated within a smaller range with decreasing in the drain spacing. The soil salinity of CK and W30 appeared accumulated throughout the whole year.

(2) The soil salinity leaching percentage of each drip irrigation under the condition of subsurface drainage was between 10% and 25% significantly higher than the value obtained from no drainage ($p < 0.05$).

(3) The soil salinity leaching curves accurately described the soil salinity leaching pattern of drip irrigation with the determination coefficients $R^2$ higher than 0.93. The soil salinity leaching curves were inversely used to determine the theoretical drip irrigation leaching quota for saline soils with different soil salinization degrees, indicating that the subsurface drainage leaching effect is clearly related to the soil salinity; for soils with higher salinity, the drip irrigation leaching efficiency was better at the same quota. Reducing the drain spacing can decrease the drip irrigation quota under the condition of the same leaching efficiency and achieve more water-saving.

(4) The groundwater table was shallow with large fluctuations under the condition of CK. The subsurface drainage reduced the groundwater table significantly ($p < 0.05$) and suppressed water level fluctuation. The reduction of drain spacing resulted in a lower groundwater table, less water level fluctuation, and the influence of the groundwater table fluctuation in no drainage area reduced.

**Author Contributions:** Conceptualization, D.L.; data curation, Y.Y.; formal analysis, Y.Y. and X.D.; funding acquisition, D.L. and X.W.; investigation, D.L.; methodology, X.Z.; project administration, X.W.; resources, Z.L.; supervision, X.W.; validation, W.H.; writing—original draft, Y.Y.; writing—review and editing, W.H. and X.W. All authors have read and agreed to the published version of the manuscript.

**Funding:** This work was supported by the Bingtuan Science and Technology Program (2021AA003), Tarim University (TDZKCQ202002), and the National Natural Science Foundation of China (51709266).

**Institutional Review Board Statement:** Not applicable.

**Informed Consent Statement:** Not applicable.

**Data Availability Statement:** All the data used in this study can be requested by email to the first author Yuhui Yang at 120130022@taru.edu.cn.

**Conflicts of Interest:** The authors declare no conflict of interest.

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
