# Peer review of "Effects of Subsurface Drainage on Soil Salinity and Groundwater Table in Drip Irrigated Cotton Fields in Oasis Regions of Tarim Basin"

_agriculture, doi:10.3390/agriculture12122167_

Round 1

Reviewer 1 Report

Review report on the paper titled “Effect of subsurface drainage on soil salinity and groundwater table in drip irrigated cotton fields in oasis region of Tarim Basin”

Overall, the paper is well written, although the manuscript needs an English language check. The topic of the research (the effect of drainage on leachate salinity) is a well-known problem in this field and investigated by many researchers. It is not clear from the arguments presented by the authors how their research contributes to any novelty in solving the said problem. The authors need to clarify this.

The introduction section must be improved with inclusion of more literature founding the background and appropriateness of the research.  The experiments conducted is described sufficiently, however, some more details are needed (which are indicated in the appended manuscript).

Author Response

Response to Reviewer 1 Comments

Please note that all line numbers in the response indicate the locations in the revised submission with tracked changes.

Point 1: Overall, the paper is well written, although the manuscript needs an English language check.

Response 1: Thank you. We have carefully revised and polished the grammar, spelling, and punctuation throughout the manuscript. Please check the revised manuscript.

Point 2: The topic of the research (the effect of drainage on leachate salinity) is a well-known problem in this field and investigated by many researchers. It is not clear from the arguments presented by the authors how their research contributes to any novelty in solving the said problem. The authors need to clarify this. The introduction section must be improved with inclusion of more literature founding the background and appropriateness of the research.

Response 2: We have improved the research background and appropriateness in the introduction, and added the following contents: “The above research results confirm that the subsurface drainage can reduce the degree of soil salinization by lowering the groundwater table and drip irrigation leaching. However, when overirrigation happens, it would increase the burden of subsurface drainage and enhance soil nutrient leaching [27]. When the drip irrigation is not enough, it could not achieve the expected soil leaching effect. Therefore, when film-mulched drip irrigations encounter shallow groundwater levels, the groundwater table distribution patterns and the soil salinity leaching characteristics of subsurface drainage farmlands need further investigation. When the drip irrigation is used for leaching under the condition of subsurface drainage, the accurate prediction of irrigation amount needs to develop appropriate calculation methods.” Please check L67-75 in Introduction part.

Point 3: The experiments conducted is described sufficiently, however, some more details are needed (which are indicated in the appended manuscript).

Response 3: We have made the following modifications to the comments in the attached manuscript.

1. “soil leching desalination rate” in L26 of abstract has been changed to “soil desalination rate”.

2. “Soil occurs salinization easily when the farmland groundwater is shallow [23].”in L58 of introduction hasbeen changed to “Soils become salinized easily when the farmland groundwater levels are shallow [23]”.

3. The comment of Table 1. “please mention the test methods of these parameters. Also, were the parameters measured by this research?”

Answer: We have introduced the measurement methods of soil bulk density, saturated soil moisture, field water-holding capacity, and permeability coefficient in 2.4. Data collection.

4. Please mention the reference for all the tests conducted, and “dry residue method”need ref.

Answer: We have added references to test methods in the text, please check L160, L161, and L170 in the revised manuscript.

Reviewer 2 Report

The topic and research idea are rather interesting. The results are indeed promising as well as exciting. However, there are some minor spelling/grammatical errors in the manuscript. The authors of this manuscript are supposed to check the manuscript and make the necessary correction in order to increase the readability of the manuscript. Please consider using the following terminology in the paper and try to amend the missing points pointed out following:

1.  Instead of "soil salt", please try to use "soil salinity" 

2. "Soil occurs salinization easily when the farmland groundwater is shallow [23]." is not clear. Please make it clear by revising the sentence.

3. The cation of "Table 1" may be revised as "Soil physical properties of the experimental field." and units should be re-checked. 

4. The meaning of ETo in "Fig. 2." should be clearly elucidated in the text.  The reader wants to know if  ETo is evapotranspiration or not. 

5. Electrical conductivity of irrigation water and its determination methodology should be given somewhere in the text and some literature such as  

[1].  Cetin, M.; Kirda, C., 2003. Spatial and Temporal Changes of Soil Salinity in a Cotton Field Irrigated with Low-quality Water. Journal of Hydrology (272):238–249. DOI:10.1016/s0022-1694(02)00268-8

[2]. Cetin, M., Ibrikci, H., Kirda, C., Kaman, H., Karnez, E., Ryan, J., Topcu, S., Oztekin, M. E., Dingil, M., Sesveren, S., 2012. Using an electromagnetic sensor combined with geographic information systems to monitor soil salinity in an area of southern Turkey irrigated with drainage water. Fresenius Environmental Bulletin (FEB), Vol 21, No 5: 1133 – 1145. 

[3]. Aragüés, R., Urdanoz, V., Cetin, M., Kirda, C., Daghari, H., Ltifi, W., Lahlou, M., Douaik, A., 2011. Soil salinity related to physical soil characteristics and irrigation management in four Mediterranean irrigation districts. Agric. Water Manage. 98, 959–966.  https://doi.org/10.1016/j.agwat.2011.01.004 

 the authors are free to cite any other references related to the topic. I believe this reference is very closely related to the topic of the manuscript.

4. Use "drain spacing" instead of "pipe spacing".

5. C0/(g/kg) in Table 7 (please check the other column headings)  should be reworded as C0 (g/kg).

6. In the last paragraph of the "discussion" section,  the sentence "The groundwater quality is poor in the test area (the salinity is about 5.0 g/L),  and the groundwater table in no drainage area is affected by the surrounding hydrological conditions (Fig. 1),  such as shallow burial and large fluctuation, and thus resulting in serious soil salinization." should be reworded as "The groundwater quality is poor in the test area (the salinity is about 5.0 g/L),  and the groundwater table in no drainage area is affected by the surrounding hydrological conditions (Fig. 1) 

Author Response

Response to Reviewer 2 Comments

Please note that all line numbers in the response indicate the locations in the revised submission with tracked changes.

The topic and research idea are rather interesting. The results are indeed promising as well as exciting. However, there are some minor spelling/grammatical errors in the manuscript. The authors of this manuscript are supposed to check the manuscript and make the necessary correction in order to increase the readability of the manuscript. Please consider using the following terminology in the paper and try to amend the missing points pointed out following:

Point 1: Instead of “soil salt”, please try to use “soil salinity”.

Response 1: Every "soil salt" in the full text has been changed to "soil salinity". Please check the revised manuscript.

Point 2: “Soil occurs salinization easily when the farmland groundwater is shallow [23].” is not clear. Please make it clear by revising the sentence.

Response 2: We have revised this sentence to “Soils become salinized easily when the farmland groundwater levels are shallow [23].” Please see L58-59.

Point 3: The cation of “Table 1” may be revised as “Soil physical properties of the experimental field.” and units should be re-checked.

Response 3: We have revised the caption as suggested and re-checked all units. Please see Table 1.

Point 4: The meaning of ET0 in “Fig. 2.” should be clearly elucidated in the text. The reader wants to know if ET0 is evapotranspiration or not.

Response 4: Done. Please see Fig. 2.

Point 5: Electrical conductivity of irrigation water and its determination methodology should be given somewhere in the text and some literature such as

[1]. Cetin, M.; Kirda, C., 2003. Spatial and Temporal Changes of Soil Salinity in a Cotton Field Irrigated with Low-quality Water. Journal of Hydrology (272):238–249. DOI:10.1016/s0022-1694(02)00268-8.

[2]. Cetin, M., Ibrikci, H., Kirda, C., Kaman, H., Karnez, E., Ryan, J., Topcu, S., Oztekin, M. E., Dingil, M., Sesveren, S., 2012. Using an electromagnetic sensor combined with geographic information systems to monitor soil salinity in an area of southern Turkey irrigated with drainage water. Fresenius Environmental Bulletin (FEB), Vol 21, No 5: 1133 – 1145.

[3]. Aragüés, R., Urdanoz, V., Cetin, M., Kirda, C., Daghari, H., Ltifi, W., Lahlou, M., Douaik, A., 2011. Soil salinity related to physical soil characteristics and irrigation management in four Mediterranean irrigation districts. Agric. Water Manage. 98, 959–966. https://doi.org/10.1016/j.agwat.2011.01.004.

the authors are free to cite any other references related to the topic. I believe this reference is very closely related to the topic of the manuscript.

Response 5: Done,please see L146 and ref. 28.

Point 6: Use “drain spacing” instead of “pipe spacing”.

Response 6: Done.

Point 7: C0/(g/kg) in Table 7 (please check the other column headings) should be reworded as C0 (g/kg).

Response 7: Corrected. Please see Table 7.

Point 8: In the last paragraph of the “discussion” section, the sentence “The groundwater quality is poor in the test area (the salinity is about 5.0 g/L), and the groundwater table in no drainage area is affected by the surrounding hydrological conditions (Fig. 1), such as shallow burial and large fluctuation, and thus resulting in serious soil salinization.” should be reworded as “The groundwater quality is poor in the test area (the salinity is about 5.0 g/L), and the groundwater table in no drainage area is affected by the surrounding hydrological conditions (Fig. 1).”

Response 8: Corrected. Please see L375-376.
